# The Effects of Massage Guns on Performance and Recovery: A Systematic Review

**DOI:** 10.3390/jfmk8030138

**Published:** 2023-09-18

**Authors:** Ricardo Maia Ferreira, Rafael Silva, Pedro Vigário, Pedro Nunes Martins, Filipe Casanova, Ricardo Jorge Fernandes, António Rodrigues Sampaio

**Affiliations:** 1Polytechnic Institute of Maia, N2i, Physical Fitness, Sports and Exercise Department, Avenida Carlos de Oliveira Campos, 4475-690 Maia, Portugal; rferreira@ipmaia.pt (R.M.F.); a037103@ipmaia.pt (R.S.); d011993@ipmaia.pt (P.V.); pmartins@ipmaia.pt (P.N.M.); 2Polytechnic Institute of Coimbra, Coimbra Health School, Physioterapy Department, Rua 5 de Outubro, São Martinho do Bispo, 3046-854 Coimbra, Portugal; 3Polytechnic Institute of Castelo Branco, Dr. Lopes Dias Health School, Physioterapy Department, Avenida do Empresário, 6000-767 Castelo Branco, Portugal; 4Sport Physical Activity and Health Research & Innovation Center (SPRINT), 4960-320 Melgaço, Portugal; 5Center for Research in Sport, Physical Education, Exercise and Health (CIDEFES), Lusófona University, 1749-024 Lisboa, Portugal; p5661@ulusofona.pt; 6Faculty of Sport of the University of Porto, Center for Research, Training, Innovation and Intervention in Sport (CIFI2D), Porto Biomechanics Laboratory (LABIOMEP), Rua Dr. Plácido da Costa, 4200-450 Porto, Portugal; ricfer@fade.up.pt; 7Sports Department, Avenida Carlos de Oliveira Campos, University of Maia, 4475-690 Maia, Portugal

**Keywords:** massage gun, performance, recovery

## Abstract

The use of massage guns has become increasingly popular in recent years. Although their use is more and more common, both in a clinical and sports context, there is still little information to guide the practitioners. This systematic review aimed to determine the effects of massage guns in healthy and unhealthy populations as pre- and post-activity or part of a treatment. Data sources used were PubMed, PEDro, Scopus, SPORTDiscus, Web of Science and Google Scholar, and the study eligibility criteria were based on “healthy and unhealthy individuals”, “massage guns”, “pre-activity, post-activity or part of a treatment” and “randomized and non-randomized studies” (P.I.C.O.S.). Initially, 281 records were screened, but only 11 could be included. Ten had a moderate risk of bias and one a high risk of bias. Massage guns could be effective in improving iliopsoas, hamstrings, triceps suralis and the posterior chain muscles’ flexibility. In strength, balance, acceleration, agility and explosive activities, it either did not have improvements or it even showed a decrease in performance. In the recovery-related outcomes, massage guns were shown to be cost-effective instruments for stiffness reduction, range of motion and strength improvements after a fatigue protocol. No differences were found in contraction time, rating of perceived exertion or lactate concentration. Massage guns can help to improve short-term range of motion, flexibility and recovery-related outcomes, but their use in strength, balance, acceleration, agility and explosive activities is not recommended.

## 1. Introduction

Since the time of the ancient Greeks, percussive or vibration therapy has been used for therapeutic and health benefits [1]. In those times, to hasten the recovery of injuries, a flexible saw was wrapped around the injured body part to transmit mechanical vibrations and improve the health in compromised individuals [1]. Progressively, with science and technology evolution, these interventions’ spectrum of use has increased, showing nowadays to be viable for various health issues (such as osteoporosis [2], chronic obstructive airway diseases [3], balance impairments [4], neurological diseases [5], acute [6] and chronic [7] musculoskeletal pain, obesity [8], vascular insufficiency [9], Down syndrome [10], bone mineral density [11], urinary stones [12], refractory acute respiratory distress syndrome [13], and enhance neuromuscular performance [14]) and patients’ (from children [15] to the elderly [16], including athletes [17] to everyday professionals [18]) conditions.

Percussive or vibration therapy could be applied with the hands (manual) or devices (mechanical) [19]. From the different hand maneuvers that could be applied, the tapotement is the most common, where the clinician applies a rapid, compressive tapping in the tissues with the edge of the hands, tips of fingers or a cupped hand, creating vibration/shock/shaking [20,21]. The development of mechanical percussion devices emerged to generate similar or greater effects to those found in manual percussion, by impacting the tissues at a different frequency, amplitude and force with higher friction, to reduce therapists’ upper extremity stress and cover larger areas in shorter treatment periods [22]. The first device was created in the 1950s [22], and nowadays, there is a wider variety to choose from, such as platforms [23], wearable devices [24], belts [25], foam rollers [26], nonportable devices [27], cushions [28] and, more recently, massage guns. 

Massage guns are hand-held mechanical devices that have a shape like a small jackhammer, are electric or battery powered and utilize different shaped applicator tips (e.g., large and small ball, flat tip, bullet/pointy tip and fork) [20] (Figure 1). These devices use percussive therapy, achieved by the rapid tip movement, to deliver bursts of pressure/vibration/massage to the myofascial tissues (i.e., fascia, muscle belly or tendon), which is thought to promote blood flow, reduce myofascial restriction and tension, improve range of motion, alleviate pain and break up trigger points [20]. As they are very versatile, massage guns have become increasingly popular in recent years, being used in both clinical and sporting contexts, for pre-activity (warm-up), post-activity (recovery) or part of a treatment [20]. 

Currently, there is still no widely accepted agreement on massage gun development, as several commercial manufacturers produce a variety of models with diverse shapes, sizes and settings (i.e., speeds/frequencies (17–53 Hz), amplitudes and applicator tips) [20]. Additionally, despite their widespread application, there is no consensus in the parameters for using most of the percussive mechanical devices. For example, it was found that, for local vibration, the frequency ranged from 5–300 Hz, the amplitude 0.12–12 mm and duration from 6 s–30 min [29]. The same was found for massage guns where, in a survey, the clinicians reported more frequently a ranged speed of 17–40 Hz, a treatment time of 30–180 s and a cadence of 2–10 s [20]. 

Despite their novelty and increasing interest by the clinical and scientific communities, there is limited scientific evidence to support their use and standardize their effects and parameters. For example, as this technology is still subject of ongoing research, many clinicians who use massage guns report anecdotal information (collaboration with peers or empirical evidence) as their main source, using them often without a specific speed, treatment time or cadence [20]. This is worrying because it goes against the principles of the optimal clinical practice and patient management [30]. Since there is an urgency to guide clinicians and understand this technology in further detail, and there is still no systematic review published related to this specific theme, the aim of the current study was to systematically review the effects of massage guns in healthy and unhealthy individuals.

## 2. Materials and Methods

This review was conducted according to the preferred reporting items for systematic reviews and meta-analyses (PRISMA) statement [31] (checklist presented in Appendix A). The review protocol was registered prospectively at the International Prospective Register of Systematic Reviews (PROSPERO—www.crd.york.ac.uk/prospero) under identification number CRD42023398894.

### 2.1. Search Strategy and Information Sources

The literature search aimed to identify studies that evaluated the effects of massage guns defined as portable hand-held mechanical devices (electric or battery powered) that utilize different shaped applicator tips to deliver bursts of percussion/vibration to the myofascial tissues. In January 2023, systematic and comprehensive searches were conducted in the PubMed, PEDro, Scopus, SPORTDiscus, Web of Science and Google Scholar electronic databases (grey literature). The search strategy was guided using the following patients, intervention, comparison, outcomes, studies (P.I.C.O.S.) model: healthy and unhealthy individuals; massage guns; other intervention, placebo sham or no intervention; pre-activity, post-activity, or part of a treatment; randomized and nonrandomized studies.

For the search strategy, a conjunction of keywords, mesh terms and established search filters were used. The main keywords used to search in the databases were “percussive therapy”, “vibration therapy” and “massage gun”. The terms (and their associates/derivatives) were then combined with the appropriate truncation and Boolean connectors. They were identified after preliminary literature searches and by crosschecking them against previous relevant studies. There was no language or publication date restrictions. Additional publications that were not found during the original database search were identified through manual searches of the personal, related studies, website bibliographies and references lists. An example of an online search strategy draft used in the MEDLINE database is presented in Figure 2:

### 2.2. Study Selection Process

Two independent authors performed the search in the electronic databases and screened the studies’ titles and abstracts to evaluate if they meet the established eligible criteria. The studies that seem to meet the criteria were gathered in EndNote and the duplicates removed using the automated software command “find duplicates”. Beyond this process, all the studies were manually checked to confirm that no duplicates remained. The preliminary included studies’ full versions were retrieved and evaluated for the fulfillment of the inclusion and exclusion criteria. The authors of the studies in which either full versions were not accessible or data were missing, were contacted by email for their access. The study selection process was supervised, and the disagreements solved through verbal discussion or arbitration by a third reviewer. The inclusion and exclusion criteria applied to this review are described in Table 1.

### 2.3. Data Extraction and Syntheses

One author performed the data collection and extraction, and to increase consistency, the process was checked by another author. The selected study-associated documents (i.e., full document, supplementary material, appendices and journal publications) were collected for analysis. The data that were extracted from the selected publications included titles, authors’ names, years of publication, instruments used, participants’ sample sizes and their characteristics, objectives, descriptions of the interventions, descriptions of the control groups, studies’ outcomes, assessment times, studies’ results and studies’ conclusions. An Excel spreadsheet was created for a proper data analysis.

### 2.4. Outcomes

Studies were combined using the most adequate qualitative and quantitative evidence synthesis. Considering the broad scope of outcomes, it was decided to restrict the work to specific umbrella terms:Primary Outcomes:
Muscular activation, strength, power;Speed, endurance, oxygen uptake;Agility, reaction, balance;Flexibility, range of motion, myofascial release;Kinetics, kinematics;Blood flow, lymphatic flow;Biomarkers of fatigue, recovery, pain, exercise-induced muscle damage, delayed-onset muscle soreness.
Secondary Outcomes:
Adverse effects after using massage gun.


### 2.5. Risk of Bias Assessment

For the risk of bias, two authors independently judged the studies, while a third author arbitrated when needed. According to the methodology, the studies were evaluated using Cochrane’s risk of bias tool, version 2 (RoB 2) or Cochrane’s risk of bias in nonrandomized studies of interventions (ROBINS-I). The randomized studies were evaluated using the five RoB 2 domains [32]: randomization process, deviations from intended interventions (intention-to-treat analysis), missing outcome data, measurement of the outcome and selection of the reported result. The nonrandomized studies were evaluated using the seven ROBINS-I domains [33]: confounding, selection of the participants, classification of interventions, deviations from intended interventions, missing data, measurement of outcomes and selection of the reported result.

## 3. Results

### 3.1. Studies’ Selection

Database searches returned 8586 records of which 8305 were duplicates. From those, 281 records were screened, but only 11 could be included [34,35,36,37,38,39,40,41,42,43,44]. The flow diagram in Figure 3 summarizes the selection process.

### 3.2. Risk of Bias

After the selection of the studies, the reviewers independently appraised the risk of bias of the 11 studies [34,35,36,37,38,39,40,41,42,43,44]. Nine [34,35,36,38,39,40,42,43] were accessed using RoB 2 and three [37,41,44] using ROBINS-I. The RoB 2 showed two problematical domains within the studies, the randomization process and the measurement of the outcome, both with the most evaluations of moderate risk of bias (100% and 75%, respectively). The domain deviations from intended interventions, missing outcome data and selection of the reported result were evaluated in all studies with low risk of bias (Figure 4a,b). In the ROBINS-I, three noticed domains were found, namely confounding, selection of the participants and selection of the reported result (one low risk of bias, seven moderate risks of bias and one serious risk of bias). The classification of interventions, deviations from intended interventions, missing data and measurement of outcomes were the best domains classified, all with low risk of bias (Figure 4c,d). The overall classification of the 11 studies was mostly of moderate risk of bias (Figure 4b,d).

### 3.3. Studies’ Synthesis

Most of the studies were performed in Europe (55%) (Austria [35], Poland [40], Portugal [43] and Spain [34,38,39]), followed by North America (27%) (USA [37,41,44]) and Asia (18%) (China [36,42]). No study was performed in Africa, Oceania or South America. Nine studies were published in peer-reviewed journals [34,35,36,38,39,40,41,42,44], whereas two were academic theses (bachelor [43] and master [37] degrees). Funding sources were reported in three studies [35,40,42] and unreported in four [37,41,43,44], with four studies reporting no funding [34,36,38,39]. One study reported a potential conflict [41] and two unreported it [37,44], while eight studies had no conflicts of interest [34,35,36,38,39,40,42,43]. The studies were published between 2020 and 2022, with the majority in 2022 (55%) [36,38,40,41,42,44]. Most study designs were crossovers [35,38,40,43] and RCTs [34,36,39,42] (both ~35%), with no treatment/placebo/sham (~25%) [35,36,38,39,40,42,43] and foam roller (~15%) [34,36] being the most frequent comparator groups. 

The outcomes explored in the studies were related to performance (~65%) [35,36,37,39,40,43,44] and recovery (~35%) [34,38,41,42], with strength (33%) [35,36,37,39,40,41,44], range of motion (19%) [35,37,41,44] and fatigue (14%) [34,38,42] being the most assessed. The massage guns were applied in the upper (28%) [39,41,42] and lower (72%) [34,35,36,37,38,40,43,44] body segments, where gastrocnemius (20%) [34,35,36,37,43,44] was the most intervened, followed by the hamstring (17%) [36,37,38,43,44] and quadricep (13%) [36,37,38,44] muscle groups. The most used massage gun was Theragun^®^ (36%) [34,37,39,44], followed by Hypervolt^®^ (18%) [35,43] and other trademarks (such as Backpack^®^ [38], Malatec^®^ [40], OUTSO^®^ [36], Rongtai^®^ [42] and TimTam^®^ [41]). From the different shaped applicator tips, the ball shape was used more often (55%) [37,38,40,41,43,44]. The frequency applied ranged from 60 Hz [36] to 20 Hz [40], with 53 Hz [35,38,43] and 40 Hz [37,39,44] (both ~20%) being the most common. The intervention times ranged from 15 s [39] to 5 min [41,42] per muscular group (2 min [34,38,43] more frequent) and 2 min [40] to 16 min [43] for the overall session (5 min [35,41] more frequent). 

Across the 11 studies [34,35,36,37,38,39,40,41,42,43,44], 281 participants were enrolled, with an average of 26 (maximum = 55 [41]; minimum = 11 [40]) per study. From those, 65 were women (23%; maximum = 26 [41]; minimum = 0 [35,36,39,43]) and 216 were men (77%; maximum = 38 [34]; minimum = 8 [40]). Per study, the average age was 24.3 ± 1.9 years (maximum = 27.2 [35]; minimum = 20.4 [36]), the height was 174.4 ± 4.1 cm (maximum = 181 [36]; minimum = 169 [37]), the weight was 72.5 ± 5.7 kg (maximum = 79.4 [35]; minimum = 57.5 [42]) and the BMI was 24.2 ± 1 kg/m^2^ (maximum = 25.6 [41]; minimum = 21.9 [36]). Table 2 provides a detailed summary of the included studies’ characteristics. 

## 4. Discussion

To proper discuss and understand the results found, the discussion section is divided into performance, recovery, physiological mechanisms, practical orientations and adverse effects and contraindications main themes.

### 4.1. Performance

Regarding the performance-related studies [35,36,37,39,40,43,44], massage guns achieved mixed results, and among the different outcomes explored, these devices appear to be effective in improving ROM and flexibility [35,37,43,44]. After a 2 min and 50 s at 40 Hz massage gun application, improvements were found (*p* < 0.001) in the Thomas test (+5.4–4.5°), 90–90 (4.8–3.4°) and ankle lunge (+4.3–3.9°) in comparison with the baseline data [37,44]. Interestingly, no differences were found in the rectus femoris [37,44]. Similarly, Konrad et al. [35] found significant statistical improvements (*p* = 0.002) in ankle dorsiflexion (+5.4°) in comparison with passive rest. Additionally, 2 min of massage gun per muscular group was found to be more effective than sham ultrasound (*p* = 0.000) and global postural reeducation (*p* = 0.02) in posterior chain muscles flexibility (seat-and-reach) [43], highlighting the positive influence that massage guns can have on these outcomes. 

In contrast, massage guns appear to be, overall, ineffective in improving strength. In explosive activities (either CMJ or DJ), the studies did not find athletic performance increments [36,37,40,44]. In fact, one study found a decrease in the jump height (*p* = 0.136; ∆ = −3.1%) after 1 min at 20 Hz massage gun application in the Achilles tendons [40]. An explanation for this result may be the massage gun’s short-term effects in stiffness reduction. As the stiffness of a muscle increases, the more motor units of the muscle are activated [45], and the more stiffness a tendon has, the more “spring-like” behavior it will have [46]. Thus, as Szymczyk et al. [40] found a stiffness reduction (∆ = −7.8∓3.6%), it was expected that the athletic performance also decreased. Only Wang et al. [36] found, in the explosive athletic performance outcomes (reactive strength index), a significant statistical improvement (*p* < 0.05) in comparison with control (no intervention). In the other athletic performance outcomes (i.e., hexagon test, lateral acceleration and Y-balance), no differences (*p* > 0.05) were found between the two groups [36]. In addition, no differences (*p* > 0.05) were found in all outcomes measured between vibration foam rollers and massage guns [36], indicating that vibration may have a greater importance on the outcomes than pressure. 

It was also found that 2 min and 30 s at 53 Hz of massage gun intervention did not influence (even having the worst score) the maximum voluntary torque in comparison with no intervention (+0.53 Nm; +0.003%; r = 0.17; *p* = 0.99 and +1.69 Nm; +1.0%; r = 0.31; *p* = 0.65, respectively) [35]. Related to these results and although no statistically significant differences (*p* > 0.05) were found in speed, power and effort index, one study revealed that those who received intervention with the massage gun performed a greater total number of repetitions (44.6 ± 4.8 vs. 39.5 ± 6.8; *p* = 0.047; ES = 0.867) and an identical number of repetitions between sets (set 1: 11.4 ± 1.2; set 2: 11.8 ± 1.1; set 3: 11 ± 1.1; set 4: 10.3 ± 2.2), compared to the control group, thus indicating a consistency of force production at resistance training [39]. However, these results may be a response to the possible positive effects that vibration guns can have in terms of recovery, as explored in more detail in the section below. As a take-home message, massage guns appear to be effective in improving ROM and flexibility, and ineffective in improving strength, balance, acceleration, agility and explosive activities.

### 4.2. Recovery

In the studies [34,38,41,42] that explored the umbrella term “recovery”, it seems that massage guns can be effective in the overall outcomes, especially those more performance related. Trainer et al. [41] found a significant statistical improvement in shoulder internal rotation ROM (+2.3°; *p* = 0.021) and strength (+1.1 lbs; *p* = 0.011) after 5 min at a 46.6 Hz massage gun application. Similarly, significant statistical increases of the upper trapezius maximum voluntary contraction were also found after a 30, 60 and 90 s fatigue protocol, with a 5 min application at either 36 or 46 Hz, in comparison with the control group (passive rest) [42]. However, the results were more consistent for the lower frequency, especially in the shorter fatigue protocols (30 and 60 s) [42]. These results may be due, once again, to the effects of massage guns in muscular stiffness reduction, as it was found, using radial displacement with tensiomyography, that massage guns at low frequency (29 Hz) had better results than manual therapy, mechanical vibration and foam rollers 24 h after a gastrocnemius eccentric fatigue protocol [34]. Moreover, the authors found trivial improvements in the contraction time (fatigue) in comparison with the other groups [34]. 

These poor results in the fatigue outcomes were evidenced in other study [38], where the authors did not find statistically significant differences in either rating of perceived exertion or lactate concentration in the massage guns group (8 min, 53 Hz) in comparison with the control group (8 min passive rest) after a 100 m water rescue. However, raw data showed some improvements after the massage gun application, as for example, the intervention group had a 9.6% decrease in the lactate and the control group only 8.1% [38]. Although small and nonstatistically significant, it is still a difference that deserves our attention. A factor that might explain the results in this study may be the frequency used. As discussed, the overall results were more consistent for lower frequencies and the Alonso-Calvete et al. [38] study used the highest of all (53 Hz), possibly affecting the positive expected results. It is also notable that the massage gun’s positive results were only found in the short term (post-intervention or 24 h) [34,38,41,42]. At longer times (48 h forward), either the studies did not show statistically significant results [34] or did not explore them [38,41,42]. Therefore, the massage guns’ long-term effects in recovery are still unknown. In summary, massage guns seem to be effective in improving short-term recovery-related outcomes.

### 4.3. Physiological Mechanisms

By its novelty, the physiological mechanisms responsible for the massage guns’ effects are not completely understood. However, taking current evidence and other similar vibrating devices into account, the physiological responses are likely due to three main categories: neuronal, vascular and mechanical.

#### 4.3.1. Neuronal

The application of mechanical vibration and pressure stimulus targets afferent cutaneous and myofascial mechanoreceptors (such as the Merkel receptors, Meissner corpuscles, Ruffini cylinders and Vater–Pacinian corpuscles), producing different neurophysiological effects [1,20]. The effect of vibration depends on stimulus provided by specific treatment parameters (duration and frequency). It has already been studied that the body has “natural frequencies” [47]. For example, the triceps surae, quadriceps and tibialis anterior range from 10 (relaxed condition) to 50 Hz (fully active state) [48]. Any interruption of these states could lead to changes in the neurophysiological functions [47]. If applied briefly at high frequencies, vibration may cause a motor unit activation, synchronization between muscle spindles, reflexive recruitment and excitation of previously inactive motor units, which ultimately leads to enhanced force production [1,14,17,36,49,50,51,52,53]. This phenomenon has been referred to as “tonic vibration reflex” [54]. However, if applied for prolonged periods with low frequencies, a decrease in muscular tonus and pain may occur via autogenic inhibition (stimulation of the Golgi tendon organs via Ib pathways) and by the gate-control theory (excitation of Aα and Aβ fibers that leads to an inhibition of the nociceptive input C fibers) [17,49,52,53,55,56]. It is also expected that a reciprocal inhibition could occur, after vibration application, through Ia inhibitory interneurons, causing relaxation of the antagonist muscle group [1,57,58]. However, massage guns are commonly used in one muscle/muscular group, and the effects are expected to be in that chosen area. So, by their use and goals, this is not a very explored physiological response. 

Another important variable could be the individuals’ levels of activity, as well-trained athletes have high muscle strength, motor neuron excitability, reflex sensitivity and fast-twitch fiber recruitment, which may lead to more elevated vibration frequency applications compared with untrained individuals in order to obtain the same results [59]. Additionally, it seems that vibration could have a more important role in the neuronal response than the pressure. In two studies [60,61] comparing nonvibrating with vibrating foam rollers, the pain perception and tolerance were greater in the vibrating group. Also, the neurophysiological changes could be beyond local and can be systemic, as an activation of somatosensory cortex, motor cortex, premotor cortex and sensorimotor cortex was found after vibration [62,63].

#### 4.3.2. Vascular

Regarding the vascular alterations, research has shown that the shear stress created by vibration may alter the nitric oxide [64], acetylcholine [65] and prostaglandin [66] concentrations levels, resulting in an increase in blood flow [9,67]. With these alterations, a prosperous environment for enhancing performance and recovery is expected to be created (e.g., cellular waste products removal, inflammation decrease, oxygen delivery increase and tissue repair/healing enhancements) [1,26]. Nevertheless, it seems that the duration of the massage guns’ application and frequency may have impact on the blood flow. One study [68] showed that massage guns improved the volume of blood flow with higher frequencies and longer durations protocols. Specifically, for 38 Hz at 5 and 10 min of 24 and 32%, respectively, and for 47 Hz at 5 and 10 min of 31 and 47%, respectively, an increase was found. Although the peak of these conditions was in the 1–3 min time range, alterations were still found in volume and velocity at 19 min [68]. An interesting finding was that these changes were due to local vibration stimulation and not excitation of the cardiovascular system, as only 4/26 subjects demonstrated minimal heart rate increases between 1 and 3 bpm, and only 5/26 subjects changed the popliteal artery diameter over 2.5% [68]. Additionally, it seems that the effect of duration may have a smaller impact than frequency. In the study mentioned above [68], the 30 Hz frequency was not sufficient to change (*p* > 0.05) the blood flow significantly, and the slopes for the linear recovery were comparable between the 5 and 10 min conditions within each frequency (47 Hz—1.7 and 1.6%/min for 10 and 5 min; 38 Hz—1.6 and 1.3%/min for 10 and 5 min; and 30 Hz—1.1 and 0.9%/min for 10 and 5 min, respectively). Similar results were found in other studies where the 30 Hz frequency was not enough to promote an increase in blood flow [69], and between 40 and 50 Hz, the mean difference in blood flow was greater for 50 Hz (although no statistical differences between them were found) [70].

In contrast, with whole-body vibration, it seems that lower frequencies (5–25 Hz) produced a greater observed effect than higher frequencies (30–50 Hz) on peripheral blood flow increase [71]. One possible explanation for these differences is that the increased blood flow may be influenced by the rate of muscle contraction. In whole-body vibration applications, lower frequencies may provide increased time between concentric and eccentric muscle-contraction cycles, allowing for greater perfusion, as higher frequencies may not allow for this perfusion, resulting in lower blood flow [71]. The vibration tonic reflexes may increase muscle metabolic demand and oxygen consumption with rhythmic contraction and relaxation of the precapillary sphincters and subsequent vasodilation [66]. Thus, it can be deduced that massage gun interventions may not be powerful enough to induce such co-contractions, relying only on the direct vibration vascular effects.

#### 4.3.3. Mechanical

Other physiological responses, partly associated with the vascular process, are the mechanical effects. It is expected that vibration and pressure may reduce muscle tension and alter the connective tissues viscoelasticity, thixotropic properties and overall mobility by an increase in fluids and temperature in the intervened area; as well, the mechanical stress could break and remobilize the scar tissue [26,43,51,72]. Between pressure and vibration, it seems that, in the massage gun devices, the vibration characteristic may have a more important role than pressure because the pressure applied by the device might not be enough to alter the tissues’ states, as happened with other techniques, such as foam rollers [38]. Supporting this statement, as shown, differences were not found between vibration foam rollers and massage guns on athletic performance [36]. 

It has been shown that vibration has the possibility to alter the tissues mechanically, as the waves can travel across the muscle’s fibers [73]. Using shear-wave elastography, it has been shown that vibration is more effective in reducing muscular stiffness compared to passive rest [74]. Again, it seems that the massage gun frequency used may have an important role when trying to achieve these goals. For example, at 30 Hz, the frequency does not seem to be sufficient to alter the fascia thickness, although an increase in temperature and a decrease in the echointensity and perceived stiffness were found [72]. However, these data were from healthy individuals, so the ceiling effect could influence the results. With other samples (e.g., chronic pain or fascia injury) [75] or with higher frequencies [35,37,41,43,44], a more evident alteration is expected. 

### 4.4. Practical Orientations

Although the information is scarce, some practical orientation can begin to be established. For recovery proposes, the evidence points to an application for prolonged periods (more than 2 min per muscular group) with low frequencies (less than 40 Hz), in order to decrease muscular stiffness and other delayed onset muscle soreness-related outcomes. For ROM and flexibility improvements, massage guns should be applied briefly (2 min or less per muscular group) at high frequencies (more than 40 Hz). It is not recommended to use massage guns immediately before (less than 5 min) a strength activity (especially an explosive one) because it can have performance-harming effects. Regarding specific application of devices, it is suggested to apply with gentle pressure (4–6/10 in visual analogic scale), dynamically (at moderate cadence), with the ball-shaped applicator tip, over the indicated area. An area requiring increased treatment can be identified by experiencing a different tactile or auditory sensation [49]. The device will either “thud” and “bounce” more aggressively over affected tissue, or practitioners may even hear a different pitch in volume from the typical vibratory sound [49]. This sound alteration should be avoided, with the practitioner having the responsibility to adjust the pressure throughout the intervention. This ensures that the vibratory application and perceived pain/discomfort are constant throughout the intervention, possibly enhancing its positive effects. Before the application, a broad patient inspection should be performed in order to understand if any contraindication is present that could inhibit it from being used (see section below).

### 4.5. Adverse Effects and Contraindications

While massage guns are generally considered safe, there are some contraindications or situations where their use may not be recommended [49,59,76,77,78,79,80]: (1) recent scars, open wounds, sunburns, rashes, bruises, bleeding or skin infections, as this can further damage the affected tissue(s) and increase the risk of infection; (2) recent fractures or bone chronic conditions (such as osteoporosis or rheumatoid arthritis), as the percussive force can interfere with the healing process or increase the risk of fracture; (3) deep-vein thrombosis or blood clotting disorders, as the pressure might dislodge blood clots and cause serious health complications; (4) diabetes and neuralgias, as it can result in numbness or loss of sensation, limiting the detection of further injury; (5) avoid using on sensitive areas of the body, such as the face, eyes, ears, head, neck, chest, spine, superficial nerves and vessels, or surgery/joint replacement (plates, metal pins, corneal or cochlear), as this can lead to serious injury and pain; (6) do not use repeatedly and aggressively, on the same area, for long periods of time (e.g., >30 min), as it can lead to muscle fiber damage, blood vessel dissections and internal bleeding (such as intra-muscular, hemartrosis, hemothorax or rhabdomyolysis); (7) for some medical conditions, its use can also be limited, such as pregnant women, fibromyalgia, migraines, hernias, hypertension, epilepsy, cancer/tumor, seizures or individuals with an implanted medications or medical devices (such as a pacemaker).

## 5. Limitations and Future Directions

One limitation of this systematic review is the moderate quality and small number of included studies. Although we understand that the main reason may be due to the device’s novelty, we expected to find more than 11 studies in the databases to be included, as massage guns are almost considered a “trendy” gadget. Add to this the fact that we consider that only 10 studies were found, since the studies from Alvarado et al. [44] and Hernandez et al. [37] belong to the same authors and were closely similar. Moreover, from the 11 studies, only short-term outcomes were evaluated, meaning the massage guns’ long-term effects remain to be understood. Also, the participants’ ages were too homogeneous. It would have been interesting to observe the effects in either injured or older samples. Additionally, the information about massage gun usage was limited. In vibration, the report of some important variables, such as frequency (number of cycles of oscillation per s—Hz), displacement (oscillatory motion—mm), acceleration (determines the magnitude—m/s^2^ or g) and duration (exposure time—min or s) is suggested [1]. Although not specific for massage guns, it would help to understand some of the results if many of them were reported. Moreover, it is also important to include information of the number and discrimination of muscles intervened in, the duration of overall and specific muscular intervention, the type of application (static or dynamic (if chosen, with the pace) and vertical or lateral (if chosen, with the angle)), the applicator tip shape used, rest periods (if sets are chosen) and other variables that may be considered as important (e.g., the tolerated pain level during intervention, person or body segment position, adverse effects and trademark). Several of these variables were not fully explored in the included studies. In the future, we endorse improved reporting and performing more quality studies, especially with other group characteristics (ages, injuries and interventions) and/or studying the physiological effects of the massage guns, as currently they are not truly understood.

## 6. Conclusions

The results of this study suggest that massage guns could be applied in order to improve short-term ROM, flexibility and recovery-related outcomes. In strength, balance, acceleration, agility and explosive activities, they either did not have improvements or they even showed a decrease in performance. Although the positive responses may be mainly mechanical, vascular and neuronal, there are still uncertainties about the physiological effects of these instruments. It is suggested that further high-quality studies need to be conducted with other population characteristics, with other comparator groups and with other intervention parameters (time and frequency) that are focused on the true physiological effects of these instruments.

## Figures and Tables

**Figure 1 jfmk-08-00138-f001:**
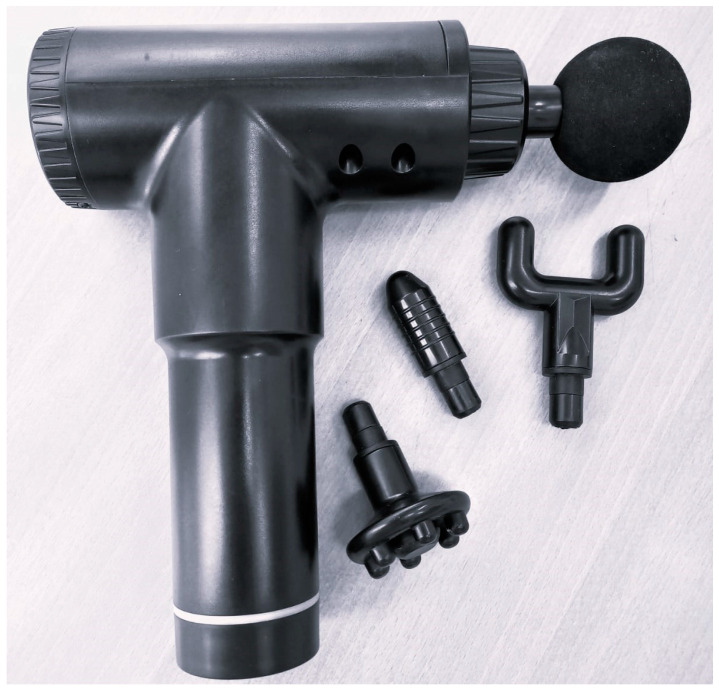
Example of a massage gun with the different attachment tips.

**Figure 2 jfmk-08-00138-f002:**
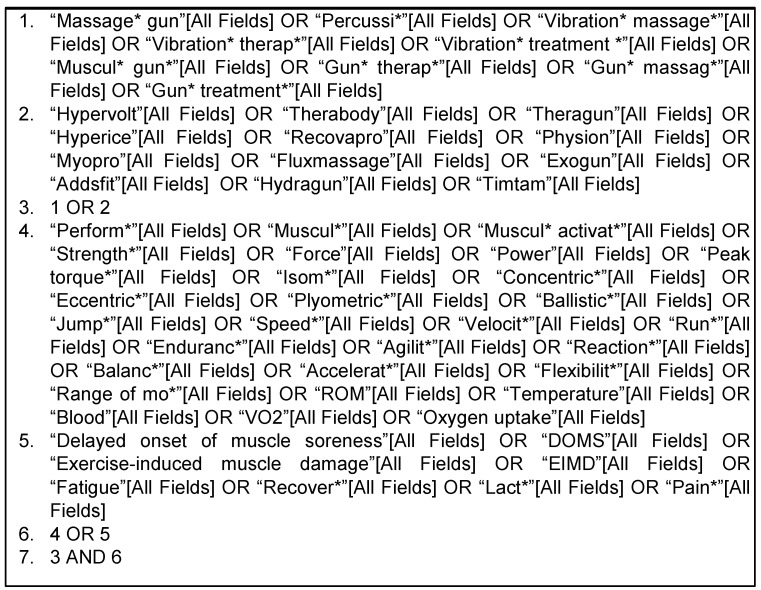
Description of the online search strategy.

**Figure 3 jfmk-08-00138-f003:**
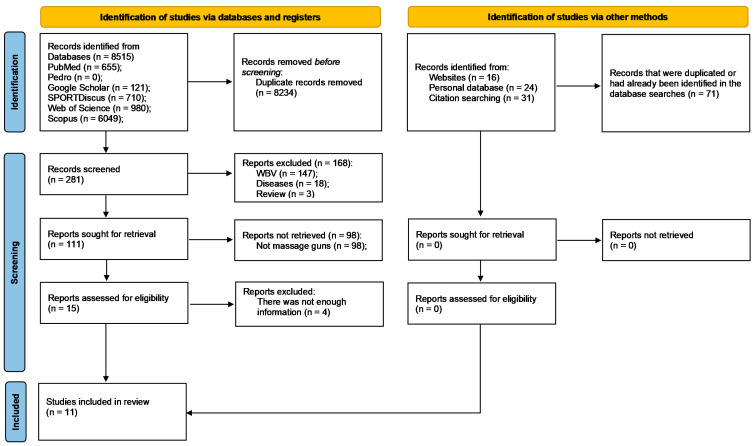
PRISMA flow diagram highlighting the selection process for the studies included in the systematic review. Abbreviations: WBV—whole body vibration.

**Figure 4 jfmk-08-00138-f004:**
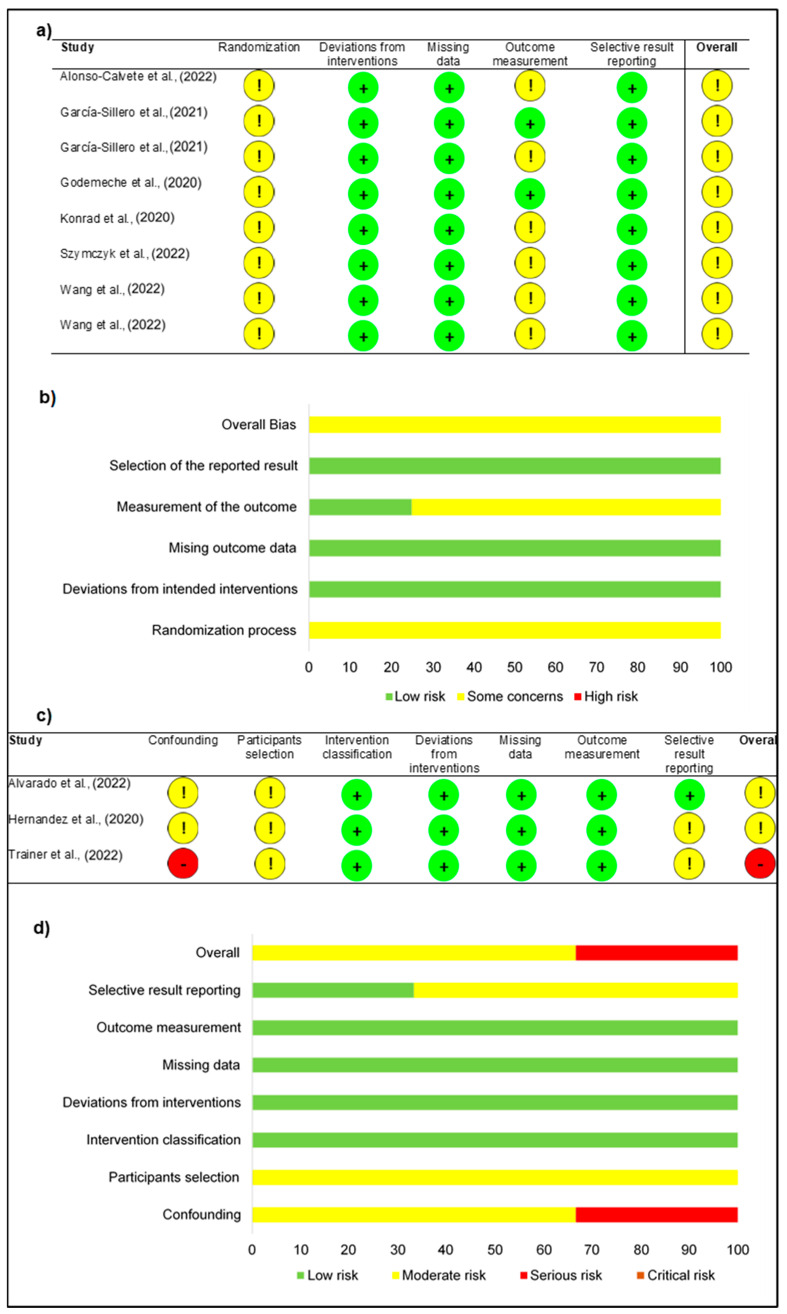
Risk of bias: (**a**) randomized trials studies assessment; (**b**) percentage distribution in randomized trials; (**c**) nonrandomized trials studies assessment; (**d**) percentage distribution in nonrandomized trials. [34,35,36,37,38,39,40,41,42,43,44].

**Table 1 jfmk-08-00138-t001:** Inclusion and exclusion criteria.

Inclusion	Exclusion
The studies must:	The studies must not:
be randomized and nonrandomized experimental studies	be books, systematic reviews, case reports, expert opinions, observational, interviews or surveys
have experimental or control groups with detailed description of the massage guns used methodology	include studies focused only on other devices or hand percussive/vibration interventions
include healthy and unhealthy (with acute injuries) individuals	include chronic injuries, illnesses, syndromes or other similar conditions
measure outcomes related with performance, injury prevention and health promotion	perform experimental or control groups composed by animals, cadaveric, in vitro or in silico
have at least one of the keywords	
be published before January 2023	
have their full version	

**Table 2 jfmk-08-00138-t002:** Results of individual studies.

Authors (A to Z)	Objectives	Participants’ Characteristics	Cohorts	Outcomes	Results
Alonso-Calvete et al. [38]	Analyze the effects of percussive massage therapy on lifeguards’ recovery after a water rescue, in comparison with passive recovery	14 lifeguards 13 male; 1 female age: 21.7 ± 2.0 years weight: 72.9 ± 11.7 kg height: 175.2 ± 9.5 cm BMI: 23.6 ± 2.1 kg/m^2^	Percussive therapy—2 min (8 min total); frequency 53 Hz; pressure 6/10 Passive recovery—8 min sitting	Fatigue (RPE; and blood lactate)	There were no differences between percussive therapy and passive recovery in lactate clearance (*p* > 0.05), finding in both modalities a small but not significant decrease in blood lactate. In perceived fatigue, both methods decreased this variable significantly (*p* < 0.001), with no significant differences between them (*p* > 0.05).
Alvarado et al. [44]	Assess the effects of percussive therapy treatment on jump performance and passive range of motion	26 recreationally active college students 15 male; 11 female age: 25.5 ± 2.5 years weight: 71.1 ± 14.9 kg height: 169.0 ± 10.0 cm	Percussive therapy—30 s (2 min and 50 s total); frequency 40 Hz	Strength (drop and countermovement jumps) Kinetic and kinematic (cameras with motion capture system) ROM (Thomas test; 90–90 hamstring; rectus femoris and ankle lunge)	A significant decrease (*p* = 0.018–0.008) in peak ankle eversion (2.0° during landing and 2.4° during take-off) was found during the drop jump following the use of percussion therapy. All other frontal/sagittal plane peak joint angle and moment changes were nonsignificant (*p* > 0.05). Moreover, percussion therapy improved ROM measures: Thomas test (*p* ≤ 0.001), 90–90 hamstring (*p* ≤ 0.001), and ankle lunge (*p* ≤ 0.001). No significance (*p* > 0.05) was found on the rectus femoris ROM and drop and countermovement jump heights.
García-Sillero et al. [34]	Compare the effects of various recovery techniques on muscle tissues after eccentric exercise-induced muscle fatigue	40 college athletes 38 male; 2 female age: 24.3 ± 2.6 years weight: 77.45 ± 8.3 kg height: 177.0 ± 6.4 cm BMI: 24.66 ± 1.6 kg/m^2^	Massage—15 min massage Vibration platform—1 min; frequency 40 Hz Percussive therapy—2 min; frequency 29 Hz Foam roller—2 sets of 30 s	Contraction time and radial displacement (tensiomyography)	The application of the different recovery techniques had positive effects for contraction time and radial displacement in the treated leg compared to the untreated leg (F = 50.01, *p* < 0.01, η^2^p = 0.58 and F = 27.58, *p* < 0.01, η^2^p = 0.43, respectively) and for the interaction of the factors (Time x Leg x Therapy: F = 5.76, *p* < 0.01, η^2^p = 0.32 and F = 5.93, *p* < 0.01, η^2^p = 0.33, respectively). The results of the various methods used were similar: contraction time (F = 0.17, *p* = 0.917; η^2^p = 0.01) and radial displacement (F = 3.30, *p* = 0.031, η^2^p = 0.22).
García-Sillero et al. [39]	Verify whether the application of percussion therapy during inter-set rest periods increases the number of repetitions during a bench press exercise	24 university students 24 male age: 24.3 ± 1.3 years weight: 77.5 ± 8.3 kg height: 177.0 ± 5.6 cm BMI: 24.7 ± 2.6 kg/m^2^	Percussive therapy—15 s (30 s total); frequency 29 Hz Control—No intervention	Strength (REPs; mean and peak velocity and peak power) Fatigue (effort index)	The percussive therapy performed a greater total number of repetitions compared to control (44.6 ± 4.8 vs. 39.5 ± 6.8; *p* = 0.047; ES = 0.867). No differences were observed for the different movement velocity, peak power and fatigue variables (*p* > 0.05).
Godemeche et al. [43]	Analyze the effectiveness of vibration massage on the flexibility of posterior chain muscles (lower limbs and lumbar spine) in active and very active	25 university students 25 male; age: 24.4 ± 0.7 years weight: 75.5 ± 2.0 kg height: 178.0 ± 1.0 cm BMI: 23.7 ± 0.5 kg/m^2^	Percussive therapy—2 min (16 min total); frequency 33.33 Hz (30 s) + 43.33 Hz (30 s) + 53.33 Hz (1 min) Global postural reeducation—“frog on the air”, 16 min Control—sham ultrasound, 16 min	Flexibility (seat and reach) Physical activity level (IPAQ)	Percussive therapy and global postural reeducation showed improvement in the posterior chain flexibility (*p* < 0.001). When comparing the two techniques, percussive therapy differs from global postural reeducation in the very active group of individuals (*p* = 0.020). In the active group, the flexibility improvements were similar in both techniques (*p* = 0.169). Both techniques were superior to the control group (*p* < 0.000).
Hernandez et al. [37]	Access the effects of myofascial release on athletic performance and passive ROM	20 university students 10 male; 10 female age: 25.5 ± 2.5 years weight: 71.1 ± 14.9 kg height: 169.0 ± 10.0 cm	Percussive therapy—30 s (2 min and 50 s total); frequency 40 Hz	Strength (drop and countermovement jumps) Kinetic and kinematic (cameras with motion capture system) ROM (Thomas test; 90–90 hamstring; rectus femoris; and ankle lunge)	In the strength, kinetic and kinematic measures of the drop and countermovement jumps, no pre-post significant differences were found (*p* > 0.05). Moreover, percussion therapy improved ROM measures: Thomas test (*p* ≤ 0.001), 90–90 hamstring (*p* = 0.001), and ankle lunge (*p* ≤ 0.001). No significance was found on the rectus femoris ROM (*p* = 0.399)
Konrad et al. [35]	Investigate the effects of a 5 min percussion treatment of the calf muscles on ROM and MVC torque of the plantar flexor muscles	16 healthy volunteers 16 male age: 27.2 ± 4.2 years weight: 79.4 ± 9.1 kg height: 179.0 ± 5.0 cm	Percussive therapy—5 min; frequency 53 Hz; 20 s cadence Passive recovery—5 min sitting	Strength (dynamometer) ROM (dynamometer)	Maximum dorsiflexion ROM increased with a large magnitude following the massage treatment by 5.4° (+18.4%; *p* = 0.002; d = 1.36), while there was no change in the control group (+1.6°; +5,3%; *p* = 0.18; d = 0.51). Moreover, torque did not change following both the percussive therapy and the control groups (*p* > 0.05).
Szymczyk et al. [40]	Investigate the impact of mechanical percussion in the Achilles tendon passive stiffness and kinematics	11 physically active 8 male; 3 female age: 24 ± 1 years weight: 68.2 ± 5.1 kg height: 170.5 ± 3.8 cm	Percussive therapy—1 min (2 min total); frequency 20 Hz Control—5 min rest	Strength (countermovement jump) Stiffness (hand-held myometer)	There were no statistically significant differences in contact time (*p* = 0.786), reactive strength index (*p* = 0.914), and relative peak power (*p* = 0.896). However, statistically significant differences in peak velocity (*p* = 0.046) and jump height (*p* = 0.03) were found. Despite that, there were no significant post hoc comparisons for jump height; it slightly decreased 5 min post-percussive therapy (*p* = 0.136; ES = −0.25; ∆ = −3.1%) compared with the control condition (*p* = 1.00; ES = 0.11; ∆ = +1.5%). There were no statistically significant differences in dominant (*p* = 0.073) and nondominant limbs’ (*p* = 0.091) Achilles tendon stiffness. Although not significant, numerically, the dominant limb Achilles tendon (*p* = 0.126; ES = −0.64; ∆ = −7.8%) had a larger reduction in stiffness immediately post-percussive therapy compared with the nondominant limb (*p* = 0.294; ES = −0.26; ∆ = −3.6%).
Trainer et al. [41]	Compare the acute effects of percussion therapy on ROM and tissue-specific measures pennation angle and muscle thickness on the dominant arm posterior rotator cuff between individuals responding positively and negatively to percussive therapy	55 healthy individuals 29 male; 26 female age: 23.9 ± 2.5 years weight: 74.9 ± 12.6 kg height: 171.1 ± 7.1 cm	Percussive therapy—5 min; frequency 46.66 Hz	ROM (digital inclinometer) Strength (dynamometer) Muscle architecture (ultrasound)	The positive response group had greater improvements than the negative response group in dominant arm internal rotation ROM (2.3° positive vs. −1.3° negative, *p* = 0.021) and internal rotation strength (1.1 lbs vs. −1.2 lbs, *p* = 0.011) after percussive therapy. No differences in external rotation strength or ROM were observed between groups (*p* > 0.05). Regarding muscle architecture, the positive group had a lesser change in teres minor muscle thickness (0.00 mm vs. 0.11 mm, *p* = 0.019) after percussive therapy. All other muscle architecture changes were not statistically different between groups (*p* > 0.05).
Wang et al., [36]	The effects of 2 different (36 Hz and 46 Hz) percussive therapy levels on upper trapezius muscles under 3 different fatigue conditions	23 healthy individuals 11 male; 12 female age: 26.5 ± 3.9 years weight: 57.5 ± 1.5 kg height: 170.5 ± 1.6 cm BMI: 24.3 ± 1.6 kg/m^2^	Percussive therapy (level 1)—5 min total; frequency 36 Hz Percussive therapy (level 3)—5 min total; frequency 46 Hz Control—no intervention	Fatigue (sEMG)	After using the vibration massage at 36 Hz, the MVC percentage of the right upper trapezius showed reductions in the 30 s, the 60 s and the 90 s fatigue task (R1: *p* = 0.022, R2: *p* = 0.005, R3: *p* = 0.049). After using the vibration massage at 46 Hz, the MVC percentage of the right upper trapezius showed a decrease in both the 60 s and the 90 s fatigue task (R2: *p* = 0.033, R3: *p* = 0.028). Significant decreases in MVC percentage for the left upper trapezius muscle were found only in the 90 s fatigue task (L3: *p* = 0.040).
Wang et al. [42]	Know if vibration foam rollers and percussion devices have an immediate impact on athletic performance during warm-up	27 tennis players 27 male age: 20.4 ± 1.3 years weight: 71.6 ± 7.8 kg height: 181.0 ± 63.0 cm	Percussive therapy—30 s (7 min total); frequency 60 Hz Vibration foam roller—30 s (7 min total); 30–40 beats per min cadence Control—no intervention	Strength (drop and countermovement jumps) Acceleration (2.5 m lateral acceleration test) Change of direction (hexagon test) Dynamic balance (Y-balance test)	In the countermovement jump, reactive strength index, and hexagon test, the difference in performance between all interventions was significant (*p* = 0.007–0.034, η^2^ = 0.266–0.364). Only those who received vibration foam roller had significantly different countermovement jump and hexagon test results when compared to the control group (53.18 ± 4.49 cm, *p* = 0.03, d = 1.26; 10.73 ± 0.4 s, *p* = 0.03, d = 1.12). Participants’ reactive strength index values were significantly different after vibration foam roller (2.01 ± 0.11 cm·mm^−1^, *p* = 0.012, d = 1.76) and percussive therapy (1.99 ± 0.11 cm·mm^−1^, *p* = 0.025, d = 1.52) compared to the control group.

Abbreviations: EMG—electromyography; IPAQ—International Physical Activity Questionnaire; MVC—maximum voluntary contraction; ROM—range of motion; REP—repetition; RPE—rating of perceived exertion.

## Data Availability

No new data were created in this study. Data sharing is not applicable to this article.

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
