# Peer review of "The Effects of Massage Guns on Performance and Recovery: A Systematic Review"

_jfmk, 2023, doi:10.3390/jfmk8030138_

Round 1

Reviewer 1 Report

Dear author,
It is meaningful to discuss the research status of electric massage guns. This could help the research go a step further to specialize in deeper phenomena. However, there is still room for improvement in the manuscript, and we look forward to improving it.
1. Topic
Please specify, what is performance? Does it refer to human behavior, movement?
2. Instructions for the massage gun
Although the author clearly stated the role of the massage gun in the text, there may be differences in the perception of the concept of the massage gun among the public. Therefore, it is recommended to place pictures (and consider the conceptual picture of the massage gun function that does not produce product marketing effects)
3. About pictures
Although the author analyzes the current survey results through the literature, if it can be presented in the form of a big data analysis map or a keyword overlap map, the visualization of the manuscript will be improved. As shown in Figure 2.
4. About Discussion
It would be helpful for the reader if the authors added a summary paragraph at the end of each discussion, presenting the main points of the chapter's analysis.
wish all the best,

Author Response

Thank you for the opportunity to revise our manuscript “The Effects of Massage Guns on Performance and Recovery: A Systematic Review”. The reviewers’ suggestions and constructive comments have been very helpful and we did our best to improve the manuscript taking the recommendations into account. We have carefully considered each suggestion and respond point-by-point. The changes are marked in the text in yellow as it was requested to be easily readable for the editors and reviewers. The manuscript revisions have been approved by all the authors. We hope the revised manuscript will better suitable and we thank you for your interest in our research.

Reviewer #1

Please specify, what is performance? Does it refer to human behavior, movement?

  • In our opinion, the performance was well specified in the 2.4 section. However, if an additional description of performance needs to be addressed, please make us know.

Although the author clearly stated the role of the massage gun in the text, there may be differences in the perception of the concept of the massage gun among the public. Therefore, it is recommended to place pictures (and consider the conceptual picture of the massage gun function that does not produce product marketing effects).

  • An image was added to help contextualize.

Although the author analyzes the current survey results through the literature, if it can be presented in the form of a big data analysis map or a keyword overlap map, the visualization of the manuscript will be improved. As shown in Figure 2.

  • We are very sorry, but we do not understand what it is needed to add in our manuscript with your suggestion. If your suggestion is critical to the manuscript approval, please tell us in more detail what is needed to be performed and, if possible, with an example of a study. Additionally, it may be a translation/interpretation error, however we do not carry out a survey in our study. Similarly, just to be clear, Figure 2 is an example of the search strategy used in an electronic database.

It would be helpful for the reader if the authors added a summary paragraph at the end of each discussion, presenting the main points of the chapter's analysis.

  • The summary of the chapter analysis is intended to be present in sub-section practical orientations. However, as suggested, a summary paragraph was added in the performance and recovery discussion sections.

Best regards,

Reviewer 2 Report

Thank you for the opportunity to review your manuscript, “The Effects of Massage Guns on Performance and Recovery: A Systematic Review.”

In general terms, the article is well presented and written and may interest readers. However, the discussion section is somewhat long, with a repetition of results in the discussion.

The layout of the manuscript has change control marks and sections, such as line 471, "Patent", which should be removed.

The PICO question is presented as an image. The authors should consider presenting it as a table or text.

The exclusion criteria cannot be non-compliance with the inclusion criteria. They must be reformulated.

Figure 3 should be moved to text. It does not contribute anything as a figure.

The Flow chart has an abbreviation that is not explained "WBV".

The discussion presents a repetition of results. It should focus on synthesizing the results and discussing them without being a repetition of data.

Line 290. Research is also the "real world". I recommend avoiding this type of expression. Moreover, if the result is small and not statistically significant, it is likely to have occurred by chance and is not reproducible. It may not have the significance attributed to it.

Author Response

Thank you for the opportunity to revise our manuscript “The Effects of Massage Guns on Performance and Recovery: A Systematic Review”. The reviewers’ suggestions and constructive comments have been very helpful and we did our best to improve the manuscript taking the recommendations into account. We have carefully considered each suggestion and respond point-by-point. The changes are marked in the text in yellow as it was requested to be easily readable for the editors and reviewers. The manuscript revisions have been approved by all the authors. We hope the revised manuscript will better suitable and we thank you for your interest in our research.

Reviewer #2

The layout of the manuscript has change control marks and sections, such as line 471, "Patent", which should be removed.

  • Thank you for the observation. In fact, it was an error that occurred when we adjusted our text to the journal layout. As requested, it was changed.

The PICO question is presented as an image. The authors should consider presenting it as a table or text.

  • As suggested, the image as changed into text.

The exclusion criteria cannot be non-compliance with the inclusion criteria. They must be reformulated.

  • In this revised version, we changed the eligibility criteria to better suit your suggestion. However, if the changes were not enough, please lets us know.

Figure 3 should be moved to text. It does not contribute anything as a figure.

  • As required, we changed the text in the image to plain text.

The Flow chart has an abbreviation that is not explained "WBV".

  • Thank you for the observation. It was added the explanation of WBV.

The discussion presents a repetition of results. It should focus on synthesizing the results and discussing them without being a repetition of data.

  • Although we understand what you are referring to, we do not consider that we repeat the results in the discussion section. Our attempt in the section was to summarize and compile the most important information from each of the studies into sub-sections, understand what the similarities and differences between them, and try to explain the reasons for the results. In some cases (specifically in the performance and recovery sub-sections), there was a need to explore each one a little further, but this was only for contextualization or support purposes, not intended to repeat the results in full. Furthermore, as the journal is open access, the general public can have access to the manuscript and, as the discussion section stands, it is easier for the beginners field readers. We thank you and respect your suggestion, but we think that the discussion section is easier to understand and read as it is currently.

Line 290. Research is also the "real world". I recommend avoiding this type of expression. Moreover, if the result is small and not statistically significant, it is likely to have occurred by chance and is not reproducible. It may not have the significance attributed to it.

  • When we added the term "real world" it was just to emphasize the results previously found. However, we understand your point of view, so we've changed the text.

Best regards,

Round 2

Reviewer 1 Report

Dear author,
I am very happy to receive the revised manuscript, which I believe has improved the problems of the previous stage. It is believed that this manuscript can help make further contributions to the research field related to massage guns.
Good luck with your research,

Reviewer 2 Report

The authors have made the requested changes.

There is no need for so much repetition of data in the discussion section, although this is a minor issue.